# A One-Year Systematic Study to Assess the Microbiological Profile in Oysters from a Commercial Harvesting Area in Portugal

**DOI:** 10.3390/microorganisms11020338

**Published:** 2023-01-29

**Authors:** Inês C. Rodrigues, Nânci Santos-Ferreira, Daniela Silva, Carla Chiquelho da Silva, Ângela S. Inácio, Maria São José Nascimento, Paulo Martins da Costa

**Affiliations:** 1ICBAS-Instituto de Ciências Biomédicas Abel Salazar, Universidade do Porto, Rua de Jorge Viterbo Ferreira, 228, 4050-313 Porto, Portugal; 2KU Leuven-Department of Microbiology, Immunology and Transplantation, Rega Institute, Laboratory of Virology and Chemotherapy, B-3000 Leuven, Belgium; 3Department of Quality Control and Food Safety, Grupo Jerónimo Martins, Rua Nossa Sra. do Amparo, 4440-232 Porto, Portugal; 4CNC-Center for Neurosciences and Cell Biology, Faculty of Medicine, University of Coimbra, Rua Larga, Polo I, 3004–504 Coimbra, Portugal; 5Faculdade de Farmácia, Universidade do Porto, Rua de Jorge Viterbo Ferreira, 228, 4050-313 Porto, Portugal; 6Interdisciplinary Centre of Marine and Environmental Research (CIIMAR), Terminal de Cruzeiros do Porto, de Leixões, Av. General Norton de Matos s/n, 4450-208 Matosinhos, Portugal

**Keywords:** oyster, farming waters, *Escherichia coli*, salmonella, antimicrobial resistance, norovirus

## Abstract

As filter-feeding animals farmed in water bodies exposed to anthropogenic influences, oysters can be both useful bioremediators and high-risk foodstuffs, considering that they are typically consumed raw. Understanding the dynamic of bacterial and viral load in Pacific oyster (*Crassostrea gigas*) tissues, hemolymph, outer shell surface biofilm, and farming water is therefore of great importance for microbiological risk assessment. A one-year survey of oysters collected from a class B production area (Canal de Mira, on the Portuguese western coast) revealed that these bivalve mollusks have a good depurating capacity with regard to bacteria, as *Salmonella* spp. and viable enterococci were not detected in any oyster flesh (edible portion) samples, despite the fact that these bacteria have regularly been found in the farming waters. Furthermore, the level of *Escherichia coli* contamination was clearly below the legal limit in oysters reared in a class B area (>230–≤4600 MPN *E. coli*/100 g). On the contrary, norovirus was repeatedly detected in the digestive glands of oysters sampled in autumn, winter, and spring. However, their presence in farming waters was only detected during winter.

## 1. Introduction

As a seafood product with high nutritional value, the Pacific oyster (*Crassostrea gigas*) is farmed across the globe, being highly appreciated in the southern European markets [1,2]. This species is also the most produced oyster in Portugal, particularly in Canal de Mira [3].

Oysters are a very particular foodstuff and one of the few animal foods that are consumed whole and raw. Furthermore, adult oysters are capable of filtering approximately 200 L of water per day, retaining many bacteria and other suspended particles [4,5]. Thus, when oysters farmed in water bodies are exposed to anthropogenic influence, their bodies concentrate chemical pollutants and fecal microorganisms, some of which can constitute a risk to human health [5,6,7]. Among pathogenic microorganisms, *Vibrio* spp., norovirus (NoV), *Salmonella* spp., and *Listeria monocytogenes* (*L. monocytogenes*) are the ones most frequently associated with foodborne zoonosis outbreaks [8,9]. Other important zoonotic agents, such as hepatitis A virus (HAV), hepatitis E virus (HEV), *Escherichia coli* (*E. coli*), and *Clostridium* spp. can be associated with exposure to contaminated shellfish [10,11,12].

In order to protect consumers’ health and ensure that oysters meet strict food safety standards, rigorous controls need to be in place concerning farming and harvesting shellfish [13]. European Hygiene Regulations [14,15,16] state that shellfish business operators are responsible for ensuring that bivalve mollusks meet strict hygiene and health standards. Risk assessment and management currently rely on the classification of shellfish harvesting areas based on the results of monitoring *E. coli* in shellfish [17] as an indicator of fecal contamination. Depending on the shellfish production area classification (A, B, or C), oysters with less than 230 MPN (most probable number) of *E. coli* per 100 g of flesh and intra-valvular liquid may go to market for direct human consumption. Nevertheless, those harvested from Class B (>230–≤4600 MPN *E. coli*/100 g) may be collected and placed on the market for human consumption only after treatment in a purification center or after relaying; oysters harvested from C areas (less than 46,000 MPN of *E. coli*/100 g) must be submitted for relaying over a longer period or undergo heat treatment to eliminate pathogenic microorganisms before being sold to consumers [16].

From an ecological perspective, oysters are a keystone species in estuarine environments as reef-builders and as filter-feeders that can naturally remove pathogens from the seawater, reducing disease risk to humans and wildlife [18]. The persistence of bacteria in oyster tissues depends on their resistance to the bactericidal activity of the hemolymph [19]. Indeed, the host resident bacteria of this circulatory fluid provide health benefits to the oyster [20] this circulatory fluid can provide information pertinent to the health assessment of bivalve populations.

Despite being valuable biofilters, oysters and other bivalves are also able to discriminate and selectively feed on different foods based on shape, surface properties, and the charge and size of particles [21]. Particle discrimination may improve water quality by removing particulate organic matter, reducing the impact of these on the ecosystem [22].

This study aimed to evaluate bacterial and viral load in Pacific oyster (*C. gigas*) flesh, intra-valvular liquid, hemolymph, outer shell surface, and farming waters during a one-year survey by analyzing the total aerobic microorganisms, marine heterotrophic bacteria, *E. coli*, *Pseudomonas* spp., *Clostridium perfringens* (*C. perfringens*), coagulase-positive *Staphylococcus*, *Enterococcus* spp., *Salmonella* spp., *L. monocytogenes*, molds, yeasts, norovirus (NoV), hepatitis E virus (HEV), and hepatitis A virus (HAV). The commercial oysters included in this study were farmed in Canal de Mira, one of the leading producers on the Portuguese western coast, which receives a continuous seawater and freshwater supply, but also inland drainage and treated and untreated urban wastewater [23].

## 2. Materials and Methods

### 2.1. Sampling and Processing

Throughout a complete seasonal cycle, summer (July 2016), autumn (November 2016), winter (January 2017), and spring (May 2017), samples of 35 cultivated Pacific oysters (*C. gigas)* (with size 9–11 cm and weight 70–90 g) and the respective farming water (1 L collected three times within 60 min intervals in three different sample points) were collected from Canal de Mira (40°38′ N, 8°45′ W) (Figure 1), in the western Portuguese coast. At the time of collection, this production area was rated as class B, meaning that live oysters from this production site could only be placed on the market for human consumption after treatment in a purification center or after relaying [16].

Samples were transported within 3 h in temperature-controlled food boxes and immediately processed upon arrival at the laboratory. The oysters were thoroughly washed with sterile seawater to remove sand, mud, and slime before the measurement of their weight, length, height, and width. They were then divided into six pools, as illustrated in Figure 2. The edible content (flesh, hemolymph, and intra-valvular liquid) of five pools, composed of five oysters each, was transferred to sterile stomacher bags and homogeneously suspended in 1/10 buffered peptone water (BPW, Biokar, Allonne, France). The sixth pool (10 oysters) was used to perform microbiological analysis on the superficial biofilm (outer shell surface), intra-valvular liquid, and hemolymph; virological analysis on the digestive gland; and metabarcoding analysis on the hemolymph. The superficial biofilm was collected by washing the shells with 100 mL of BPW using a pair of sterile toothbrushes. Intra-valvular fluid was collected into a sterile falcon after filtration through a sterile gaze. The hemolymph was collected with a sterile syringe, followed by the dissection of the digestive glands.

### 2.2. Bacterial Analysis

The microbiological analysis of flesh and intra-valvular liquid samples was performed in compliance with the European Union microbiological criteria for live bivalve mollusks [17], taking also into account the potential microbiological hazards of raw oyster consumption. The analysis included the enumeration of total aerobic microorganisms at 7 °C and 30 °C, marine heterotrophic bacteria at 21 °C, *E. coli*, *Pseudomonas* spp., *Clostridium perfringens* (*C. perfringens*), coagulase-positive *Staphylococcus*, *Enterococcus* spp., molds and yeasts, and the detection of *Salmonella* spp. and *L. monocytogenes*. For bacterial enumeration, a pooled sample comprising 25 oysters was used. Regarding the detection of *Salmonella* spp. and *L. monocytogenes*, five pools comprising five oysters each were prepared. Total aerobic microorganisms at 30 °C, marine heterotrophic bacteria at 21 °C, *E. coli*, and *Enterococcus* spp. were also assessed on the superficial biofilm, intra-valvular liquid, and hemolymph samples (pool of 10 oysters). Finally, the total counts of aerobic microorganisms at 22 °C and 37 °C, marine heterotrophic bacteria at 21 °C, *E. coli*, *Enterococcus* spp., and *Salmonella* spp. were also evaluated in the farming water samples.

International Organization for Standardization (ISO) methods were used for the enumeration/detection of microorganisms: ISO 4833-1 (ISO 4833-1, 2013) and ISO 6222 (NF EN ISO 6222, 1999) for total aerobic microorganisms; ISO 16649-2 (NF ISO 16649-2, 2001) and ISO 16649-3 (ISO 16649-3, 2014) for *E. coli*; ISO 6579 (NF EN ISO 6579/A1, 2007) for *Salmonella* spp.; ISO 7937 (ISO 7937, 2004) for *C. perfringens*; ISO 6888-3 (ISO 6888-3, 2003) for coagulase-positive *Staphylococcus*; ISO 7899-2 (PN EN ISO 7899-2, 2004) for *Enterococcus* spp.; ISO 11290-1 (ISO 11290-1, 2017) for *L. monocytogenes*; and ISO 21527-2 (ISO 21527-2, 2003) for yeasts and molds (Appendix A). The detection of both *Pseudomonas* spp. and marine heterotrophic bacteria was performed using an internal laboratory method (Appendix A). Briefly, to detect marine heterotrophic bacteria, the pour-plate method was performed. In total, 1 mL of each sample was plated in marine agar medium (Condalab, Madrid, Spain) and incubated at 21 °C for 48 h. The detection of *Pseudomonas* spp. was performed using serial dilutions and spreading 100 µL of each sample on cephaloridin fucidin cetrimide (CFC) agar (Oxoid, Basingstoke, UK), and the plates were incubated at 30 °C for 48 h.

In addition, bacteria total counts on samples of superficial biofilm, intra-valvular liquid, and hemolymph collected during summer were also estimated by fluorescence in situ hybridization (FISH), as previously described by [24,25,26,27]. Eco440, PseaerA, and GV probes (MWG-Biotech, Ebersberg, Germany) were used to detect *E. coli*, *P. aeruginosa*, and *Vibrio* spp., respectively. The slides were mounted using Vectashield^®^ Mounting Medium (Vector Laboratories, Newark, CA, USA) and immediately observed in a Nikon Eclipse E400 microscope (Nikon Instruments, Amsterdam, The Netherlands) at 1000× magnification with an oil immersion objective (HCX PLAN APD). All samples were analyzed in triplicate, and the data are presented as cell/milliliter.

### 2.3. Antimicrobial Susceptibility Testing

The antimicrobial susceptibility of all *E. coli* and *Enterococcus* spp. isolated from the farming waters, and from the flesh, superficial biofilm, intra-valvular liquid, and hemolymph of oysters was tested and interpreted according to the Clinical and Laboratory Standards Institute guidelines (CLSI, 2018), using the Kirby–Bauer method. A panel of 18 and 15 antimicrobial agents was used for the antimicrobial susceptibility testing of *E. coli* and *Enterococcus* spp. strains, respectively (Table 1). All antimicrobial disks were from Oxoid (Oxoid, Basingstoke, UK). Isolates resistant to at least one antibiotic agent of three or more antibacterial classes were considered multidrug-resistant (MDR) bacteria [28].

### 2.4. Detection of Food- and Waterborne Viruses

The detection of norovirus (NoV), hepatitis E virus (HEV), and hepatitis A virus (HAV) was performed on both the farming waters and the oysters’ digestive gland samples from all seasons (Figure 2) following ISO/TS 15216-1:2017 ‘Microbiology of food and animal feed—Horizontal method for determination of hepatitis A virus and norovirus in food using real-time RT-PCR—Part 1: Method for quantification’ and as previously described. [29,30]. Briefly, viral extraction was carried out from the homogenates of each sample and mixed with 2 mL of proteinase K (0.1 mg/mL). This mixture was spiked with 10 μL of a virus used to control extraction efficiency, the murine norovirus (MNV-1; 2.7 × 10^9^ RNA copies/μL), followed by agitation for 1 h at 37 °C at 320 osc/min (ELMI DOS-10 M Digital Orbital Shaker, ELMI, Riga, Letonia). Then, it was incubated for 15 min at 60 °C and centrifuged for 5 min at 3000× *g* at room temperature. In total, 500 μL of supernatant was recovered and used for RNA extraction using an NZY Total RNA Isolation Kit (NZYTech, Lisbon, Portugal), according to the manufacturer’s instructions. RNA was eluted in 50 μL of RNA-free sterile water and stored at −80 °C until further analysis. NoV GI and GII and HAV were quantified using the primers/probes described in ISO 15216-1:2017. The detection and quantification of HEV were performed by an RTqPCR assay targeting the ORF3 region with the primers/probes previously described [30,31]. The RTqPCR assays were performed using the iTaq Universal PROBES One-Step Kit (Bio-Rad Laboratories, Hercules, CA, USA) in a final volume of 20 μL reaction mixture according to the manufacturer’s and run in a CFX Connect Real-Time System (Bio-Rad Laboratories).

The presence of oyster herpesvirus type 1 (OsHV-1) was also evaluated on oyster edible portions from all seasons (Figure 2). The DNA extraction of the oyster edible portions was performed using a QIAamp cador Pathogen Mini Kit (Qiagen, Hilden, Germany) following the manufacturer’s instructions. Briefly, 50 mg of tissue and fluids were subjected to ‘Pretreatment T2–Enzymatic Digestion of Tissue’, followed by ‘Pretreatment B1–for Difficult-to-lyse Bacteria in whole blood or Pre-treated Tissue’ and finally ‘Purification of Pathogenic Nucleic acids from Fluid Samples’. Eluted DNA was stored at −80 °C until further analysis. OsHV-1 quantification was performed following an improved protocol published by Martenot et al., using a Taqman probe and primers that target the B region of the OsHV-1 genome. qPCR was performed using SsoAdvanced Universal PROBES Supermix (Bio-Rad Laboratories) [32].

### 2.5. Metabarcoding Analysis for Microbiome Composition

DNA extraction for metabarcoding analysis was performed using a QIAamp cador Pathogen Mini Kit (Qiagen, Hilden, Germany) following the manufacturer’s instructions. For edible samples, ‘Pretreatment T2–Enzymatic Digestion of Tissue’ was used, followed by ‘Pretreatment B1–for Difficult-to-lyse Bacteria in whole blood or Pre-treated Tissue’, and for hemolymph samples, ‘Pretreatment B2–for Difficult-to-lyse Bacteria in Cell-free Fluids’ was used.

The metabarcoding analysis was carried out in edible portion samples collected in the four seasons and hemolymph samples collected in autumn and spring, using next-generation sequencing (GATC Microbiome Profiling (Combined Analysis)) (GATC Biotech, Constance, Germany). This amplicon-based method targeted the V1-V8 variable region of the 16S rRNA gene, using the primers 27F (AGAGTTTGATCCTGGCTCAG) and BS-R1407 (GACGGGCGGTGWGTRC), resulting in a fragment of 1381 bp. The data were checked for chimeras using UCHIME, and the corresponding sequences were removed from further analysis. Non-chimeric, unique sequences were then subjected to BLASTn analysis using non-redundant 16S rRNA reference sequences with an E-value cutoff 1 × 10^6^. Reference 16S rRNA sequences were obtained from the Ribosomal Database Project. Only good quality and unique 16S rRNA sequences that have a taxonomic are considered and used as a reference database to assign operational taxonomic unit (OTU) status to the sequences. Taxonomic classification was based on NCBI Taxonomy [5]—http://www.ncbi.nlm.nih.gov/taxonomy (accessed on 15 December 2017). Except for the E-value cutoff (1 × 10^6^), no other thresholds were used during the BLAST analysis. All the hits to reference the 16S rRNA database were considered, and specific filters were applied to the hits to remove false positives. Further, the best hit and multiple hits per sequence were analyzed separately to determine the discriminatory power of the sequences with respect to the assigned OTUs. Finally, the classification of OTU sequences was consolidated to compute relative abundancies (percentage composition).

## 3. Results

### 3.1. Morphological Parameters

In each season, four morphological parameters were evaluated individually: total weight, height, length, and width (Appendix A). The total weight varied between 56.9 g ± 5.0 (spring; mean body weight ± S.D.) and 84.3 g ± 18.5 (winter). Considering the total height measurements, the minimum values recorded were 2.6 cm ± 0.4 (spring), and the maximum values were 3.0 cm ± 0.3 (winter). Regarding the total length measurements, the values varied between 8.4 cm ± 0.8 (spring and summer) and 10.2 cm ± 1.6 (winter). Finally, the total width values varied between 4.7 cm ± 0.6 and 5.2 cm ± 0.5, where the highest and lowest widths were observed in the winter and spring, respectively.

### 3.2. Bacterial Analysis

In the present study, the microbiological quality of oysters and their farming waters was examined in four seasonal sampling surveys (Table 2). *Salmonella* spp. and *L. monocytogenes* were not found in the flesh or intra-valvular liquid. The level of *E. coli* contamination was found to be between 20 (summer and spring) and 92 (winter) MPN *E. coli*/100 g in the edible portion. Furthermore, viable enterococci were not detected in any flesh or intra-valvular liquid samples. On the contrary, *Salmonella* spp., *Enterococcus* spp., and *E. coli* were detected in the farming waters. *E. coli* was detected in all seasons, whereas *Enterococcus* spp. was detected in summer, autumn, and winter, and *Salmonella* spp. was only detected in summer and winter in the farming water samples. The highest concentration of heterotrophic marine bacteria in the farming water (1.5 × 10^4^ CFU/100 mL) was found in the sample collected in winter.

The number of total microorganisms on superficial biofilms (covering the outer shell) seems to have followed their abundance in the farming water, particularly in the samples collected in winter and spring. On the contrary, the number of marine heterotrophic bacteria on the surface biofilm of the oysters and their feeding waters was less articulated: whereas similar values were found in autumn, in summer and in spring, the difference exceeded two logarithms.

Regarding intra-valvular liquid samples, there were differences between the number of microorganisms detected in this physiological fluid and the quantity found in the farming water column. A higher concentration of microorganisms in the intra-valvular liquid was found compared to the farming water column during summer and autumn. Despite this, the number of fecal bacteria (*E. coli* and *Enterococci*) was generally higher in water than in the intra-valvular liquid, similarly to what was observed with the superficial biofilm. Likewise, marine heterotrophic bacteria in the intra-valvular liquid showed the same dynamics when compared to the superficial biofilm. Indeed, the highest value observed for this group of microorganisms was with the sample collected during summer. On the other hand, the hemolymph showed an increased number of marine heterotrophic bacteria in the summer and spring samples.

Analysis of the superficial biofilm, intra-valvular liquid, and hemolymph samples by the FISH protocol revealed the presence of *Vibrio* spp. in 100% of the samples (Table 3). The hemolymph was the most contaminated material (median, 4.1 × 10^6^ cells/g), and the highest values of *Vibrio* spp. cells were observed during summer. The FISH method allowed the detection of *E. coli* cells in 75% of the superficial biofilm samples (summer, winter, and spring), whereas the bacteriological method only detected viable *E. coli* in the winter sample. A clear contrast between the traditional plating method and the FISH cell counting was also observed with regard to *Pseudomonas* spp.

### 3.3. Antimicrobial Susceptibility Testing

In this study, we have evaluated the antimicrobial susceptibility of 30 *E. coli* isolates that were obtained from the farming water (n = 18), edible portion (n = 10), intra-valvular liquid (n = 1), and superficial biofilm (n = 1), and 20 *Enterococcus* spp. isolated from the farming water (n = 10), superficial biofilm (n = 9) and intra-valvular liquid (n = 1). Overall, 27% of the *E. coli* and 1% of the enterococci isolates were susceptible to all of the antimicrobial drugs tested. The remaining 22 *E. coli* and 19 *Enterococcus* spp. isolates showed resistance to at least one antimicrobial drug. The frequency of antimicrobial susceptibility to each antimicrobial drug on *E. coli* and *Enterococcus* spp. isolates was calculated and is presented in Figure 3 and Figure 4, respectively.

Regarding *E. coli* isolates, the highest rate of drug resistance was observed for ampicillin and cephalothin (approximately 25%), followed by nalidixic acid (16.7%), amoxicillin/clavulanic acid, aminoglycosides (gentamicin, tobramycin, and streptomycin), aztreonam, cefoxitin, ciprofloxacin, tetracycline, and sulfamethoxazole/trimethoprim (8.3%), and chloramphenicol and doxycycline (4.2%). Concerning *Enterococcus* spp. isolates, nitrofurantoin revealed the highest prevalence of resistance, followed by linezolid, rifampicin, and tetracycline (16.7%), ampicillin (11.1%), doxycycline and quinupristin-dalfopristin (5.6%). It is worth mentioning that neither the third-generation of cephalosporin-resistant *E. coli* nor vancomycin-resistant *Enterococcus* were found. However, the high frequency of resistance to aminopenicillins, second-generation cephalosporins, and nalidixic acid in *E. coli* isolates, and the resistance levels against ampicillin and linezolid in enterococci, deserve to be highlighted.

Moreover, 30 susceptibility profiles of *E. coli* isolates and 20 susceptibility profiles of *Enterococcus* spp. isolates were analyzed, where 37% and 35% were revealed to be MDR *E. coli* isolates and MDR *Enterococcus* spp. isolates, respectively. The resistance profiles of MDR strains are shown in Table 3 and Table 4. MDR *E. coli* was isolated from the farming water (n = 6), edible portion (n = 4), and superficial biofilm (n = 1) (Table 4), and MDR *Enterococcus* spp. was detected in the farming water (n = 6) and superficial biofilm (n = 1) (Table 5). Regarding seasonality, MDR *E. coli* was found in farming water in summer, autumn, and winter samples, and the edible content in autumn and winter samples. On the other hand, during the winter, eight samples were found to be contaminated with MDR *E. coli* in the edible portion and superficial biofilm. The water contaminated with MDR *Enterococcus* spp. was collected during summer (n = 1), autumn (n = 3), and winter (n = 3). Furthermore, the superficial biofilm containing MDR *Enterococcus* spp. was collected during winter (n = 1).

### 3.4. Detection of Food- and Waterborne Viruses

The analysis of foodborne viral contamination was performed, and the results are shown in Table 6. NoV was detected in the digestive gland in the spring and summer samples, as well as in the farming water in spring. HEV was detected in the farming water in spring. HAV was not detected in any digestive gland or water samples. OsHV-1 was also not detected in any edible portion sample.

### 3.5. Metabarcoding Analysis

Metabarcoding analysis revealed that the microbiome of the edible portion and hemolymph throughout seasons were dominated by *Vibrio* spp. (22.3%), excluding the edible portion in the winter sample (O3C) (Figure 5). The most predominant microorganisms belong to the genus *Vibrio* followed by *Psychrilyobacter* (Table 7).

## 4. Discussion

Presently, official controls to prevent food poisoning associated with raw oyster consumption are based on the classification of their harvesting areas. Oysters examined in this study were harvested on the Canal de Mira, which is under threat of organic pollution and limited water renewal as it is a long narrow inlet of the seacoast, where freshwater from the Vouga River mixes with seawater from the Atlantic Ocean [3]. Therefore, the low rainfall and the increase in tourism during summer [36] are possible contributors to the rise of heterotrophic marine bacteria and *Salmonella* contamination in the farming water. Indeed, the higher prevalence of *Salmonella* in Portugal during the summer months [37] might help its spread into the aquatic environment. On the other hand, the detection of this pathogenic bacterial species during winter is most likely due to rainfall or surface runoff [38]. Enterococci and *E. coli* monitoring confirmed that this oyster farming area is exposed to fecal pollution, although the level of contamination was not as high as expected, considering that Canal de Mira is under anthropogenic pressure and receives treated/untreated sewage discharges [39].

Regardless of the sampling period, the edible portion of oysters showed compliance with the microbiological safety criteria set out in [16,17,33,34,35]. The level of *E. coli* contamination was clearly below the legal limit for *E. coli* contamination in oysters reared in a class B area (>230–≤4600 NMP *E. coli*/100 g). However, samples collected in the rainy seasons of autumn and winter showed the highest total microorganisms, *Pseudomonas* and *C. perfringens* contamination. In the spring, the bacterium *C. perfringens* was found below the detection level of 10 CFU/g, and the MPN of *E. coli* per 100 g of flesh was the lowest and only comparable to that obtained in the summer sampling. However, the biometric measurements during this study suggested that oysters should be harvested during the winter due to their greater growth during this season. Previous work [40] found that the summer months have a negative impact on oyster growth and their immunological parameters as the oysters are exposed to high temperatures and low food availability, recovering during the autumn and winter months.

Nevertheless, taking into account only the bacteriological assessment, oysters could be harvested at any time of the year, as the microorganisms of greatest concern (*Salmonella* spp. and *L. monocytogenes*) were not detected in any of the samples collected, and fecal indicator bacteria contamination levels were also low compared to those reported by other authors [7,41,42,43]. Furthermore, *E. coli* contamination levels were clearly below the European Union legal end product standard (230 MPN/100 g) [17] and enterococci, which are broadly recognized by their resistance to environmental stress [44], were not found (<10 UFC/g) in any flesh sample included in this study.

Despite having been proven that microbial colonization of oyster outer shell is shaped by the number and nature of microorganisms present in the farming water [45,46], neither *Salmonella* spp. nor enterococci were found, and *E. coli* was only found in the winter sample. Similarly, hemolymph analysis did not show contamination with the fecal bacteria that were detected in the farming water and the intra-valvular liquid.

As filter feeders, oysters developed a highly sophisticated innate immune system that is able to recognize and eliminate various microorganisms via an array of orchestrated immune reactions [47,48]. This “depuration capacity” has been previously reported in *Anodonta cygnea* for enterococci and *E. coli* [49], and also in *C. gigas* for *Salmonella* Newport [50]. Hemolymph is pivotal in oyster immune defense, and hemocytes are the main effector cell population, capable of selectively recognizing, adsorbing, internalizing, and inactivating non-symbiotic microorganisms [51,52]. Indeed, oysters’ hemolymph is not sterile, being a rich microbial environment (10^2^–10^5^ bacteria per g) composed mainly of organisms of the genera *Vibrio*, *Pseudomonas*, *Aeromonas*, and *Alteroromas* [51].

Analysis of the hemolymph by the FISH protocol revealed the presence of *Vibrio* spp., *P. aeruginosa*, and *E. coli* cells in 100%, 75%, and 25% of the samples, respectively, whereas the bacteriological method was unable to detect any colony-forming *E. coli* in hemolymph or pseudomonas in the flesh. These contrasting data were most likely due to the presence of viable but non-culturable (VBNC) bacterial cells, which are characterized by having a better fitness for survival under stressful conditions. In 2021, Wagley et al. [53] showed that *V. parahaemolyticus* VBNC cells could be resuscitated (100% revival) under favorable conditions.

In this study, we observed that the peak of *Vibrio* spp. cells on the surface of shells and intra-valvular liquid observed in summer was most likely the result of the proliferation of this genus with warmer water temperatures [6,54,55]. Metabarcoding analysis revealed high levels of *Vibrio* spp. in both the flesh and hemolymph during summer and autumn. *Vibrio* spp. plays an important role in oyster welfare, but also in public health, as it could be either an oyster pathogen, associated with mass summer mortalities of *Crassostrea gigas* or a zoonotic pathogen, including *V. parahaemolyticus* (the principal causes of seafood-borne disease linked to the consumption of shellfish) and *V. vulnificus*, which may cause serious wound infections [54,56,57]. Moreover, this study showed that hemolymph contained more *Vibrio* spp. compared to the edible content, which could be explained by the immunological function of hemolymph. Indeed, the overall microbiome of oysters displays a seasonal influence, also mentioned by Scannes et al. (2021) [58].

This is, to our knowledge, the first study that the occurrence of norovirus (NoV), hepatitis A virus (HAV), hepatitis E virus (HEV), and oyster herpesvirus type 1 (OsHV-1) in the Canal de Mira production area. NoV and HEV were both detected, but NoV was more frequent and the only one found simultaneously in the digestive gland and water samples collected during winter. According to Lowther et al. (2012) [43], this seasonality is typical in Europe and it might be explained by the convergence of several factors: the higher prevalence of noroviruses in the human population, the greater persistence of viral particles under winter environmental conditions (low temperature and low solar irradiation), and lower viral clearance in oysters due to the slowing of the metabolism. In the present investigation, HAV and OsHV-1 were not found in any of the samples analyzed. Since 2008, OSHV-1 has been causing epidemics with high mortality in *C. gigas* throughout Europe. To the best of our knowledge, OSHV-1 has only been detected in one sample of *C. gigas* harvested in Portugal, although the authors reported that this animal could have been imported from France.

Antimicrobial resistance remains a serious global health concern, being considered one of the most pressing global issues by the World Health Organization (2020) [59]. Paradoxically, wastewater treatment can favor the emergence and spreading of antimicrobial resistance (AMR) as resident bacterial communities are exposed to sub-inhibitory concentrations of antimicrobials (due to the elimination of these substances in the feces and urine of medicated individuals), favoring the transfer of genes between bacteria and their subsequent dissemination into aquatic environments [60,61]. The consequences of these events were found in this research, as evidenced by the isolation of both *E. coli* and *Enterococcus* spp. multidrug-resistant strains and the high frequency of resistance to important classes of antimicrobial drugs.

## 5. Conclusions

The present study was performed with a limited number of samples, which may result in a misestimation of prevalence. However, this is the first report assessing a wide range of microbiological parameters of oysters and their farming waters, combining genomics and classical plating methods to both commensal and microorganisms of great concern. In common with previous studies, the contrast between the results for the presence of *E. coli* and norovirus demonstrates the limitations of using *E. coli* to estimate and manage the risk of human enteric virus in oysters.

## Figures and Tables

**Figure 1 microorganisms-11-00338-f001:**
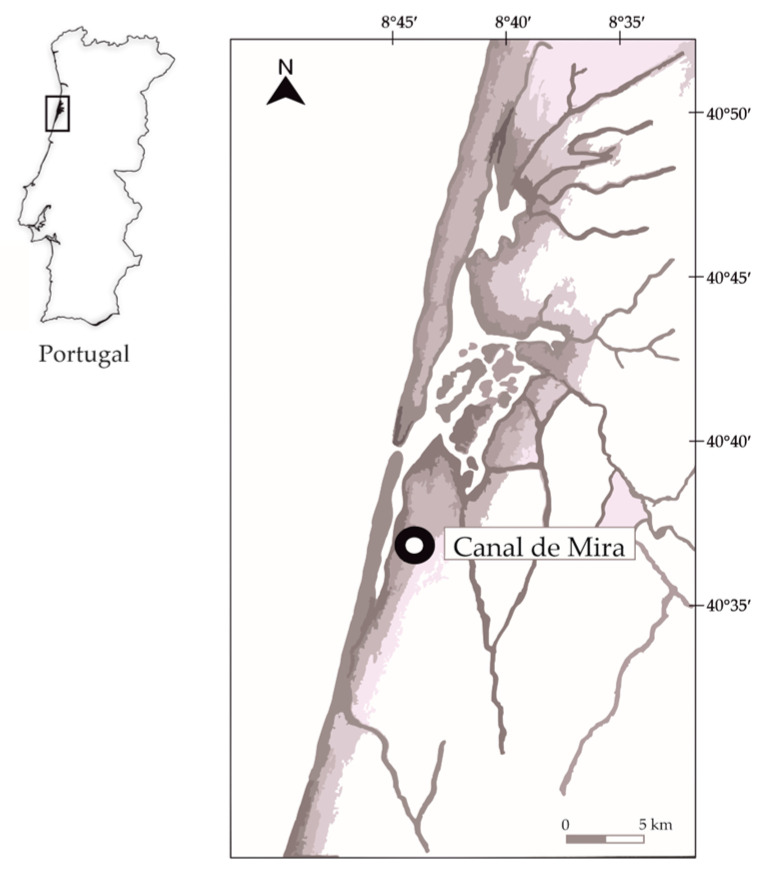
Schematic figure representing the Canal de Mira arm of Ria de Aveiro. This channel is an elongated and shallow arm, 25 km long, that runs south-southwest, parallel to the west Portuguese coastline.

**Figure 2 microorganisms-11-00338-f002:**
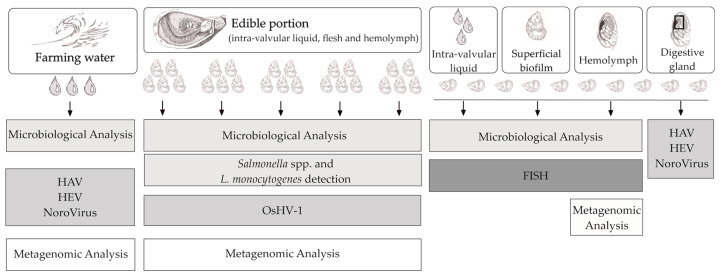
Experimental diagram. OsHV-1: oyster herpes virus; HAV: hepatitis A virus; HEV: hepatitis E virus; FISH: fluorescence in situ hybridization.

**Figure 3 microorganisms-11-00338-f003:**
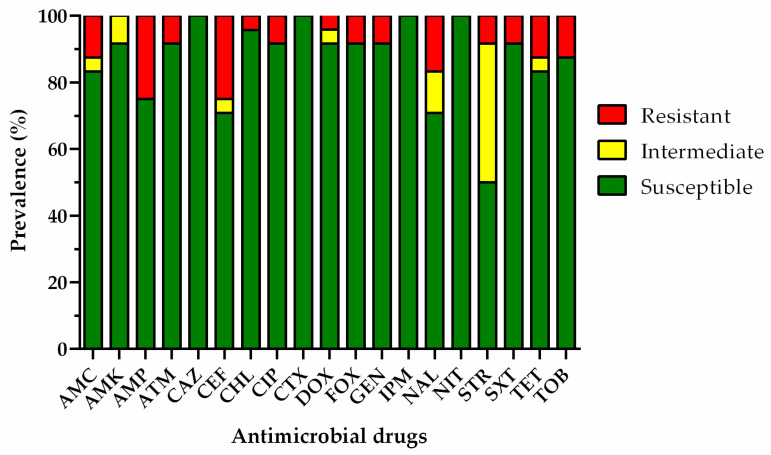
Percentage of resistance in *E. coli* isolated from the farming water and the edible portion of oysters. AMC: amoxicillin/clavulanic acid, AMK: amikacin, AMP: ampicillin, ATM: aztreonam, CAZ: ceftazidime, CEF: cephalothin CHL: chloramphenicol, CIP: ciprofloxacin, CTX: cefotaxime, DOX: doxycycline, FOX: cefoxitin, GEN: gentamicin, IPM: imipenem, NAL: nalidixic acid, NIT: nitrofurantoin, STR: streptomycin, SXT: sulfamethoxazole/trimethoprim, TET: tetracycline, TOB: tobramycin.

**Figure 4 microorganisms-11-00338-f004:**
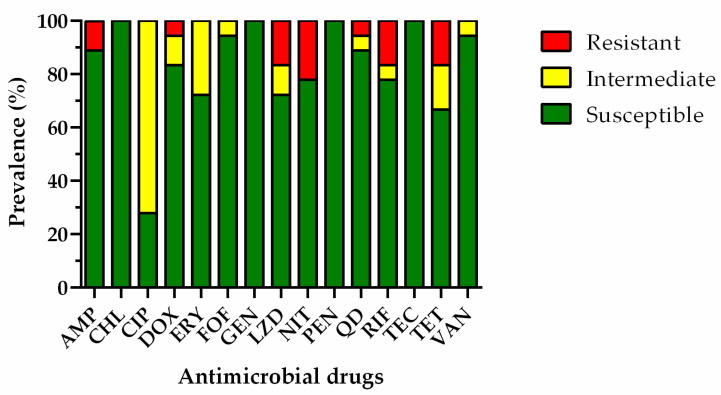
Percentage of resistance in *Enterococcus* spp. isolated from the farming water and the edible portion of oysters. AMP: ampicillin, CHL: chloramphenicol, CIP: ciprofloxacin, DOX: doxycycline, ERY: erythromycin, FOF: fosfomycin, GEN: gentamicin, LZD: linezolid, NIT: nitrofurantoin, PEN: penicillin, QD: quinupristin-dalfopristin, RIF: rifampicin, TEC: teicoplanin, TET: tetracycline, VAN: vancomycin.

**Figure 5 microorganisms-11-00338-f005:**
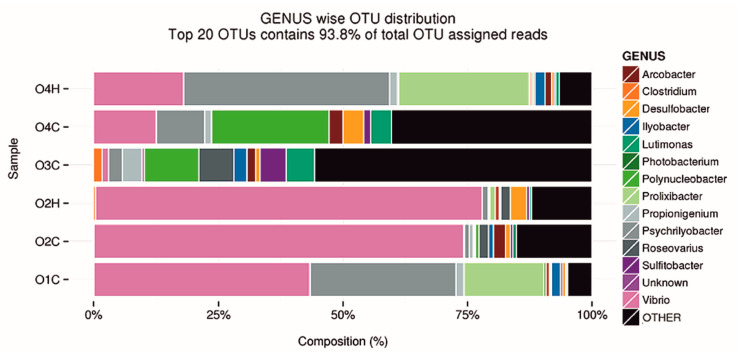
Genus distribution of microbiome of oysters. O4H: hemolymph in spring; O4C: edible portion in spring; O3C: edible portion in winter; O2H: hemolymph in autumn; O2C: edible portion in autumn; O1C: edible portion in summer.

**Table 1 microorganisms-11-00338-t001:** Antimicrobial agents used to evaluate the resistance profile of *E. coli* and *Enterococcus* spp. isolated from farming water, edible portion, hemolymph, and superficial biofilm samples.

Microorganism	Antimicrobial Agent	Acronym	Disc Content (μg)
*E. coli*	Amoxicillin/clavulanic acid	AMC	30
	Amikacin	AMK	30
	Ampicillin	AMP	10
	Aztreonam	ATM	30
	Chloramphenicol	CHL	30
	Cephalothin	CEF	30
	Cefoxitin	FOX	30
	Cefotaxime	CTX	30
	Ceftazidime	CAZ	30
	Ciprofloxacin	CIP	5
	Doxycycline	DOX	30
	Gentamicin	GEN	10
	Imipenem	IMP	10
	Nalidixic acid	NAL	30
	Streptomycin	STR	10
	Sulfamethoxazole/trimethoprim	SXT	25
	Tetracycline	TET	30
	Tobramycin	TOB	10
*Enterococcus* spp.	Ampicillin	AMP	10
	Chloramphenicol	CHL	30
	Ciprofloxacin	CIP	5
	Doxycycline	DOX	30
	Erythromycin	ERY	15
	Fosfomycin	FOF	200
	Gentamicin	GEN	10
	Nitrofurantoin	NIT	300
	Penicillin	PEN	10
	Quinupristin-dalfopristin	Q-D	15
	Rifampicin	RIF	5
	Teicoplanin	TEC	30
	Tetracycline	TET	30
	Vancomycin	VAN	30
	Linezolid	LZD	30

**Table 2 microorganisms-11-00338-t002:** Microbiological analysis of farming waters, edible portions (flesh, hemolymph, and intra-valvular liquid), superficial biofilm, intra-valvular liquid, and hemolymph during all seasons, using classic methods.

	Microorganisms		Samples			
			Summer	Autumn	Winter	Spring
Farming water (CFU/100 mL)	Total microorganisms	22 °C	2.3 × 10^3^	2.0 × 10^4^	1.5 × 10^3^	1.1 × 10^3^
		37 °C	2.8 × 10^3^	2.5 × 10^3^	8.0 × 10^2^	2.0 × 10^2^
	Marine heterotrophic bacteria		1.5 × 10^4^	3.6 × 10^3^	2.3 × 10^3^	3.2 × 10^3^
	*E. coli*		3.8 × 10^0^	1.8 × 10^1^	6.0 × 10^0^	1.0 × 10^0^
	*Salmonella* spp.		Present in 1L	Absence in 1L	Present in 1L	Absence in 1L
	*Enterococcus* spp.		3.0	9.2 × 10^1^	2.0	<1
Edible portion	Total microorganisms (CFU/g)	30 °C	4.4 × 10^2^ (S ^B^)	1.2 × 10^3^ (S ^B^)	1.2 × 10^3^ (S ^B^)	1.3 × 10^2^ (S ^B^)
		7 °C	2.5 × 10^2^ (S ^B^)	6.0 × 10^2^ (S ^B^)	8.2 × 10^2^ (S ^B^)	1.1 × 10^2^ (S ^B^)
	Marine heterotrophic bacteria (CFU/g)		9.0 × 10^3^	1.6 × 10^4^	3.8 × 10^4^	5.9 × 10^4^
	*E. coli* (MPN)/100 g)		20.0 (S ^A^)	36.0 (S ^A^)	92.0 (S ^A^)	20.0 (S ^A^)
	*Pseudomonas* spp. (CFU/g)		<100	<100	<100	<100
	*Salmonella* spp.		Absence in 25 g (S ^A^)	Absence in 25 g (S ^A^)	Absence in 25 g (S ^A^)	Absence in 25 g (S ^A^)
	*Clostridium perfringens* (CFU/g)		<10 (S ^B^)	2.0 × 10^1^ (U ^B^)	3.0 × 10^1^ (U ^B^)	<10 (S ^B^)
	coagulase + *Staphylococcus* (CFU/g)		<100 (S ^C^)	<100 (S ^C^)	<100 (S ^C^)	<100 (S ^C^)
	*Enterococcus* spp. (CFU/g)		<10	<10	<10	<10
	*Listeria monocytogenes*		Absence in 25 g S ^B^	Absence in 25 g S ^B^	Absence in 25 g S ^B^	Absence in 25 g S ^B^
	Molds (CFU/g)		<25 S ^B^	<25 S ^B^	5.0 × 10^1^ S ^B^	1.0 × 10^2^ S ^B^
	Yeasts (CFU/g)		<25 S ^B^	5.0 × 10^1^ S ^B^	<25 S ^B^	5.0 × 10^1^ S ^B^
Intra-valvular liquid (CFU/mL)	Total aerobic microorganisms at 30 °C		2.4 × 10^4^	2.0 × 10^4^	3.8 × 10^2^	6.0 × 10^2^
	Marine heterotrophic bacteria		3.1 × 10^5^	2.3 × 10^2^	5.1 × 10^3^	4.6 × 10^4^
	*E. coli*		6.7 × 10^2^	<1	<1	<1
	*Enterococcus* spp.		<1	<1	5.0	<1
Superficial biofilm (CFU/g)	Total aerobic microorganisms at 30 °C		3.6 × 10^4^	2.1 × 10^5^	2.6 × 10^3^	4.5 × 10^3^
	Marine heterotrophic bacteria		1.5 × 10^6^	2.7 × 10^3^	4.9 × 10^4^	5.1 × 10^5^
	*E. coli*		<0.1	<0.1	1.1	<0.1
	*Enterococcus* spp.		9.2 × 10^−1^	<0.1	1.9	7.3 × 10^−1^
Hemolymph (CFU/mL)	Total aerobic microorganisms at 30 °C		2.0 × 10^3^	7.6 × 10^3^	4.3 × 10^3^	1.4 × 10^2^
	Marine heterotrophic bacteria		3.3 × 10^5^	1.6 × 10	2.0 × 10^2^	3.5 × 10^5^
	*E. coli*		<1	<1	<1	<1
	*Enterococcus* spp.		<1	<1	<1	<1

S ^A^: Satisfactory according to [17]; S ^B^/U ^B^: Satisfactory/Unsatisfactory according to [33,34]; S ^C^: Satisfactory according to [35]. CFU: colony-forming unit, MPN: most probable number.

**Table 3 microorganisms-11-00338-t003:** Bacterial quantification using the FISH method on superficial biofilm, intra-valvular liquid, and hemolymph in four seasonal sampling surveys.

Microorganism	Sample	Summer	Autumn	Winter	Spring
*P. aeruginosa*	Superficial biofilm	5.0 × 10^2^	9.0 × 10^2^	1.5 × 10^3^	1.3 × 10^3^
	Intra-valvular liquid	5.0 × 10^2^	1.3 × 10^6^	4.3 × 10^3^	8.0 × 10^2^
	Hemolymph	2.0 × 10^2^	6.0 × 10^2^	6.8 × 10^3^	<100
*Vibrio* spp.	Superficial biofilm	3.9 × 10^7^	5.0 × 10^2^	4.0 × 10^2^	1.0 × 10^3^
	Intra-valvular liquid	2.3 × 10^7^	3.4 × 10^3^	7.0 × 10^2^	2.2 × 10^4^
	Hemolymph	5.0 × 10^6^	4.6 × 10^6^	2.3 × 10^3^	6.8 × 10^6^
*E. coli*	Superficial biofilm	5.0 × 10^2^	<100	2.0 × 10^2^	6.0 × 10^2^
	Intra-valvular liquid	1.3 × 10^7^	<100	<100	<100
	Hemolymph	3.2 × 10^7^	<100	<100	<100

Data are expressed as cells/mL (intra-valvular liquid and hemolymph) and as cells/g (superficial biofilm).

**Table 4 microorganisms-11-00338-t004:** Resistance patterns in *E. coli* MDR strains isolated from the farming waters, edible portions, and superficial biofilm samples.

Isolate	Season	Sample	Resistance Profile
1	Summer	Water	AMP ^R^ CEF ^I^ CIP ^R^ DOX ^I^ NAL ^R^ STR ^R^ SXT ^R^ TET ^R^
2	Autumn	Water	ATM ^R^ CEF ^R^ STR ^I^
3	Autumn	Edible portion	AMC ^R^ AMP ^R^ CEF ^R^ STR ^I^ TET ^I^
4	Winter	Edible portion	CEF ^R^ NAL ^I^ STR ^I^
5	Winter	Edible portion	CEF ^R^ GEN ^R^ NAL ^R^ STR ^I^ TET ^R^
6	Winter	Edible portion	AMC ^R^ AMP ^R^ CEF ^R^ FOX ^R^ NAL ^R^
7	Winter	Superficial biofilm	AMC ^R^ AMP ^R^ FOX ^R^
8	Winter	Water	AMC ^I^ AMP ^R^ CHL ^R^ CIP ^R^ CEF ^R^ DOX ^R^ GEN ^R^ NAL ^I^ STR ^R^ SXT ^R^ TET ^R^ TOB ^R^
9	Winter	Water	AMK ^I^ AMP ^R^ CEF ^I^ NAL ^R^ STR ^I^
10	Winter	Water	CEF ^R^ DOX ^R^ STR ^I^ TET ^R^
11	Winter	Water	DOX ^R^ NAL ^R^ STR ^R^ TET ^R^

AMC: amoxicillin/clavulanic acid, AMP: ampicillin, CEF: cephalothin, CHL: chloramphenicol, CIP: ciprofloxacin, DOX: doxycycline, FOX: cefoxitin, GEN: gentamicin, NAL: nalidixic acid, STR: streptomycin, TET: tetracycline, TOB: tobramycin, SXT: sulfamethoxazole/trimethoprim, R: resistant, I: intermediate.

**Table 5 microorganisms-11-00338-t005:** Resistance patterns in *Enterococcus* spp. MDR strains isolated from the farming water, edible portion, and superficial biofilm samples.

Isolate	Season	Sample	Resistance Profile
1	Summer	Water	CIP ^I^ DOX ^I^ ERY ^I^ RIF ^I^ TET ^I^
2	Autumn	Water	CIP ^I^ Q-D ^R^ TET ^R^
3	Autumn	Water	DOX ^R^ ERY ^I^ FOF ^I^ LZD ^R^ Q-D ^I^ RIF ^R^ TET ^I^
4	Autumn	Water	AMP ^R^ CIP ^I^ LZD ^I^
5	Winter	Water	CIP ^I^ ERY ^I^ LZD ^I^ Q-D ^R^ RIF ^R^
6	Winter	Water	AMP ^R^ CIP ^I^ ERY ^I^ RIF ^R^
7	Winter	Superficial biofilm	CIP ^I^ LZD ^R^ TET ^I^

AMP: ampicillin, CIP: ciprofloxacin, DOX: doxycycline, ERY: erythromycin, FOF: fosfomicine, LZD: linezolide, Q-D: quinupristin-dalfopristin, RIF: rifampicin, TET: tetracycline, R: resistant, I: intermediate.

**Table 6 microorganisms-11-00338-t006:** Detection of norovirus (NoV) hepatitis A virus (HAV), hepatitis E virus (HEV), and oyster herpesvirus type 1 (OsHV-1) in sampled matrix.

Season	Sample	NoV	HAV	HEV	OsHV-1
Summer	Digestive gland	ND	ND	ND	NA
	Water	NA	NA	NA	NA
	Edible portion	NA	NA	NA	ND
Autumn	Digestive gland	Detected	ND	ND	NA
	Water	ND	ND	ND	NA
	Edible portion	NA	NA	NA	ND
Winter	Digestive gland	Detected	ND	ND	NA
	Water	ND	ND	ND	NA
	Edible portion	NA	NA	NA	ND
Spring	Digestive gland	Detected	ND	ND	NA
	Water	Detected	ND	Detected	NA
	Edible portion	NA	NA	NA	ND

ND: Not Detected, NA: Not Applicable.

**Table 7 microorganisms-11-00338-t007:** Genus composition (in percentage) of edible portion and hemolymph during all seasons.

Genus	O1C	O2C	O2H	O3C	O4C	O4H
*Vibrio*	43.2%	74.2%	77.6%	1.3%	12.5%	18.0%
*Clostridium*	0.1%	0.1%	0.4%	1.8%	0.0%	0.0%
*Psychrilyobacter*	29.3%	1.0%	1.2%	2.6%	9.7%	41.3%
*Polynucleobacter*	0.1%	0.7%	0.3%	11.0%	23.6%	0.3%
*Prolixibacter*	16%	0.1%	1.1%	0.0%	0.0%	26.2%
*Desulfobacter*	0.5%	1.0%	3.3%	0.9%	4.2%	0.5%
*Arcobacter*	0.4%	2.4%	0.0%	1.8%	2.8%	1.3%

O1C: edible portion in summer, O2C: edible portion in autumn, O2H: hemolymph in autumn, O3C: edible portion in winter, O4C: edible portion in spring, O4H: hemolymph in spring.

## Data Availability

The original contributions presented in the study are included in the article. Further enquiries can be directed to the corresponding author.

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
