# Peer review of "A One-Year Systematic Study to Assess the Microbiological Profile in Oysters from a Commercial Harvesting Area in Portugal"

_microorganisms, 2023, doi:10.3390/microorganisms11020338_

Round 1
Reviewer 1 Report
the manuscript is well written and and reflects a great deal of competent microbial work making it worthy of publication. That said there are not any great revelations in the manuscript. All results are essentially confirmatory
Author Response
Reviewer 1
The manuscript is well written and reflects a great deal of competent microbial work making it worthy of publication. That said there are not any great revelations in the manuscript. All results are essentially confirmatory.
We would like to thank Referee 1 for the positive comment on the manuscript.
Reviewer 2 Report
Although the initial idea of this research is interesting and addresses to a diverse microbiological analysis, there are many obvious flaws, of which I mention a few:
1) Missing References (e.g., in line 53 the references [8-10] only refers to Norovirus, hepatitis A virus and metals present in several microorganisms, not to "... foodborne zoonosis outbreaks";
2) Definition of specific objectives (very vague, "...... a baseline analysis of the bacterial and viral load in Pacific oysters...";
3) Small number of samples studied (n=35 in one year, which limits the results and conclusions to be drawn); unclear how they were worked (how many samples were included in each pool? How were they selected?);
4) Unclear/not adequately described methods and incorrect References (for example, in the section "2.4 Detection of food- and waterborne viruses", the authors do not mention which method was used to detect OsHV-1;
5) The metagenomic analysis was performed for what purpose? On how many samples? Which programs were used in the analysis? Not clear at all;
6) Unclear way of results presentation (e.g., Table 1 presents a value for each parameter (average?) and not the values for each sample/pool studied; TSA and analysis by "Season" were only presented for E. coli and Enterococcus – Table 3 and 4; Is difficult to understand what was found by the authors;
7) Although the discussion is well structured, based on what was presented in the results, I do not agree with the authors when they state (line 344" "... This study comprises an extensive assessment of the microbial load of Pacific oysters...".
Author Response
Reviewer 2
Although the initial idea of this research is interesting and addresses to a diverse microbiological analysis, there are many obvious flaws, of which I mention a few:
- Missing References (e.g., in line 53 the references [8-10] only refers to Norovirus, hepatitis A virus and metals present in several microorganisms, not to "... foodborne zoonosis outbreaks";
We added the reference of The European Union One Health 2021 Zoonoses Report (EFSA, 2021) and the work of Potasman et al., 2002, which includes the foodborne zoonosis outbreaks caused by Norovirus, Vibrio spp. and Salmonella spp. and Listeria monocytogenes. Regarding the other microorganisms, such as Clostridium spp., E. coli, HEV and HAV mentioned in the same line, the authors have rephrased the sentence, clarifying their importance as agents present in shellfish associated with zoonoses. All the references have been amended.
- Definition of specific objectives (very vague, "...... a baseline analysis of the bacterial and viral load in Pacific oysters...";
We have now rephrased the paragraph as it as follows: “This study aimed to evaluate bacterial and viral load in Pacific oyster (C. gigas) flesh, intra-valvular liquid, hemolymph, outer shell surface and farming waters during a one-year survey by analysing the total aerobic microorganisms, marine heterotrophic bacteria, E. coli, Pseudomonas spp., Clostridium perfringens (C. perfringens), coagulase-positive Staphylococcus, Enterococcus spp., Salmonella spp., L. monocytogenes, molds, yeasts, norovirus (NoV), hepatitis E virus (HEV) and hepatitis A virus (HAV). Commercial oysters included in this study were farmed in Canal de Mira, one of the leading producers on the Portuguese western coast, which receives a continuous seawater supply, but also inland drainage and treated and untreated urban wastewater.
- Small number of samples studied (n=35 in one year, which limits the results and conclusions to be drawn); unclear how they were worked (how many samples were included in each pool? How were they selected?);
We thank the reviewer for the comment. We verified in the text that it was not clear how many oysters were included in our study. We clarified it in Figure 2 (now improved and with higher quality): “Sampling was carried out in Summer, Autumn, Winter and Spring. Overall, 140 oysters (commercial size) were randomly collected during a one-year survey, 35 in each season.”
- Unclear/not adequately described methods and incorrect References (for example, in the section "2.4 Detection of food- and waterborne viruses", the authors do not mention which method was used to detect OsHV-1;
Details on OsHV-1 detection in oyster edible portion were added in 2.4. Detection of food- and waterborne viruses section.
- The metagenomic analysis was performed for what purpose? On how many samples? Which programs were used in the analysis? Not clear at all;
In the experimental design, we included several methods to explore their virtues and limitations to assess oyster microbial “qualities”, namely their food safety. With this objective in mind, we broaden the spectrum of microbial species (bacteria, viruses, molds and yeasts) to be detected/enumerated. Moreover, we also used methods which allow the detection of viable but non-culturable bacterial cells, including lower abundant (yet biologically meaningful) genera and species. Naturally, in the discussion section, we focus on the findings that could be most relevant and interesting to readers.
A detailed explanation of metagenomic data analysis and programs used was added to the 2.5 section.
- Unclear way of results presentation (e.g., Table 1 presents a value for each parameter (average?) and not the values for each sample/pool studied; TSA and analysis by "Season" were only presented for E. coli and Enterococcus – Table 3 and 4; Is difficult to understand what was found by the authors;
We corrected this by adding additional information to the materials and methods. For bacterial enumeration, a pooled sample comprising 25 oysters was used. Regarding the detection of Salmonella spp. and L. monocytogenes, five pools comprising five oysters each were prepared. Total aerobic microorganisms at 30 °C, marine heterotrophic bacteria at 21 °C, E. coli and Enterococcus spp. were also assessed on the superficial biofilm, intra-valvular liquid and hemolymph samples (pool of 10 oysters).
Concerning the data on Tables 3 and 4, it should be stressed that given the volume of results we tried to summarize our findings by focusing on a quantitative picture of (a) the presence of viable bacteria (comprising three important bacterial genera/species) on oyster’s superficial biofilm and biological fluids, and of (b) the resistance patterns of E. coli and Enterococci to evaluate the environmental dispersion of MDR strains (some of them display resistance against critically important antimicrobials). As oysters are typically consumed raw, multidrug-resistant strains found in our study can be easily “returned” to humans.
- Although the discussion is well structured, based on what was presented in the results, I do not agree with the authors when they state (line 344" "... This study comprises an extensive assessment of the microbial load of Pacific oysters...".
We thank Referee 2 for the positive comment on the discussion. We changed the first sentence of this section: “Presently, official controls to prevent food poisoning associated with raw oyster consumption are based on the classification of their harvesting areas. Oysters examined in this study were harvested on the Canal de Mira, which is under threat of organic pollution and limited water renewal […]
Reviewer 3 Report
In the introduction - let it be clarified whether there are other similar studies (not for the region, but for Oysters).
Row 80 - specify a sample how many mussels it includes/or the sample corresponds to one mussel. Clarification needed.
It is good to give tabular information about the size and weight classes of the examined mussels (as well as the number of each size/weight class) in point "2.1. Sampling and processing".
Authors should consider using descriptive statistics when displaying the results (rows 180 to 187, as well as the other results obtained) to describe the main statistical characteristics of the sample.
Row 273 To add more detail to the discussion. The conclusion is very interesting.
Recommendations
- Improve the quality of the figures;
- From line 357 to 362. How do the authors explain these results? I recommend a better interpretation of the results;
- Is it possible to assess the risk in different accidental situations (volley pollution with wastewater, ingress of contaminated water, etc.)?
Author Response
Reviewer 3
In the introduction - let it be clarified whether there are other similar studies (not for the region, but for Oysters).
We have probably been too brief referencing similar studies because we are already dealing with an excess of bibliographical references due to the nature of the manuscript (research paper). However, references to other similar studies were plentifully inserted in the discussion section.
Row 80 - specify a sample how many mussels it includes/or the sample corresponds to one mussel. Clarification needed.
Figure 1 (now Figure 2) was reformulated to better illustrate the study design, the spectrum of samples studied, and the methods used. Additionally, more detailed information was added to the materials and methods: “For bacterial enumeration, a pooled sample comprising 25 oysters was set. Regarding the detection of Salmonella spp. and L. monocytogenes, five pools comprising five oysters each were prepared. Total aerobic microorganisms at 30 °C, marine heterotrophic bacteria at 21 °C, E. coli and Enterococcus spp. were also assessed on the superficial biofilm, intra-valvular liquid and hemolymph samples (pool of 10 oysters).”
It is good to give tabular information about the size and weight classes of the examined mussels (as well as the number of each size/weight class) in point "2.1. Sampling and processing".
In this work, only market-size oysters were used. Information regarding their weight, height, length and width was relegated to supplementary information, precisely because one reviewer found it unnecessary.
Authors should consider using descriptive statistics when displaying the results (rows 180 to 187, as well as the other results obtained) to describe the main statistical characteristics of the sample.
More information about the morphological parameters (with descriptive statistics) has been added to the supplementary material (Table S2).
Row 273 To add more detail to the discussion. The conclusion is very interesting.
Recommendations
- Improve the quality of the figures;
We agree that the resolution of Figure 1 should be improved and we have reuploaded all figures with higher quality.
- From line 357 to 362. How do the authors explain these results? I recommend a better interpretation of the results;
The presence of multidrug-resistant strains in oysters (which had no apparent exposure to antimicrobials) demonstrates that the selective effects of antimicrobials are far beyond their prescription. It is known that effluents from urban areas and animal husbandry, even when properly treated, discharge resistant bacteria (especially bacteria of faecal origin, like E. coli and enterococci) in the receiving waters. Furthermore, antibiotic-contaminated effluents (of Sewage treatment plants and pharmaceutical manufactories) facilitate the development of resistant bacteria in the aquatic environment. However, the authors consider that these explanations, although evident, transcend the experimental design of their work, so they consider it more relevant to underline what they have factually found: from a public health perspective, oysters are a vehicle of resistant bacteria for humans, and this risk is made worse by the fact that these bivalves are eaten raw. From an ecological perspective, our results confirmed that resistance, once developed, is not confined to the limits of the ecological niche where it primarily emerged.
- Is it possible to assess the risk in different accidental situations (volley pollution with wastewater, ingress of contaminated water, etc.)?
We thank the reviewer for the comment. Unfortunately, the study we conducted was not meant for that assessment. We can only hypothesize about it as a factor that contributes to the observed results. However, a study with such a purpose (although complex) would be crucial to rationally fix the location and boundaries of oysters’ production areas.
Reviewer 4 Report
The paper reported the data of microbiological profile in oysters collected from Portugal. The data may be useful for the local consumers and local officials in charge of the seafood. However, the paper is like a monitoring report, rather than a research article. This is lack of a hypothesis to test. I suggest the paper must be substantially improved to increase its novelty.
Some specific comments:
1) The resolution of figures is too low and the figure captions should be improved.
2) A map of sampling locations may be helpful.
3) I don't understand the scientific significance of the morphological parameters of oyters (section 3.1). Does it mean that you choose the similar audlt oysters? If so, the large difference in size should be avoided. If not, how can you compare the data?
Author Response
Reviewer 4
The paper reported the data of microbiological profile in oysters collected from Portugal. The data may be useful for the local consumers and local officials in charge of the seafood. However, the paper is like a monitoring report, rather than a research article. This is lack of a hypothesis to test. I suggest the paper must be substantially improved to increase its novelty.
We thank the reviewer for the comment. In fact, our one-year survey was based on oysters farmed at a specific location on the Portuguese western coast. However, our results clearly showed that current official microbiological criteria for oysters need to be revised or supplemented. We understand that the limitations of using E. coli to estimate and manage the risk of human enteric viruses in oysters is not especially novel, as it converges with other previous investigations. But we believe that we (i) describe a comprehensive set of analytical methods combining genomics and classical plating methods towards commensal bacteria, fecal indicators and pathogenic microorganisms, (ii) studied a topic that is scientifically challenging, i.e., the relationship between farming water contamination levels and bacterial and viral load on oysters, highlighting that these bivalve molluscs may have a good depurating capacity with regards to pathogenic bacteria. Of particular importance is the fact that (iii) we also quantified bacteria total counts on samples of superficial biofilm, intra-valvular liquid and hemolymph, a topic that has been an existing gap in the available literature.
In that sense, we are deeply convinced that our results have economic and public health importance for guiding a more efficient control of oyster production and for consumer health protection.
In this revised version, the authors improved the experimental diagram highlighting the diversity of methods used to characterize the dynamic of bacterial and viral load in Pacific oysters. Furthermore, the authors have added some details in Materials and Methods. These modifications are highlighted with tracking changes in the attached document.
Some specific comments:
The resolution of figures is too low and the figure captions should be improved.
We agree that the resolution of the Figures should be improved and have reloaded the figures with higher quality. Figure 2 was reformulated to better illustrate the study design, the spectrum of samples studied, and the methods used.
Please let us know if there are any additional improvements are needed that we should consider.
A map of sampling locations may be helpful.
The map has now been included.
I don't understand the scientific significance of the morphological parameters of oyters (section 3.1). Does it mean that you choose the similar adult oysters? If so, the large difference in size should be avoided. If not, how can you compare the data?
We thank the reviewer for the comment. The study we conducted was not meant to assess the influence of the morphological parameters on microbiological patterns. Indeed, only adult oysters (ready to be placed on the market for human consumption) were sampled. Therefore, the graphs were moved for “supplementary data”.
Reviewer 5 Report
Comments and Suggestions for Authors
Abstract
L. 28-29: what does Class B production area mean?
Introduction
L. 58-66: How the monitoring is done? How frequently the monitoring occurs? How many oysters are usually used for it? For the broad audience this classification has to be better explained. Is it based on NMP of E. coli? What are the limits of NMP for Class A, B and C? Additionally, change the order of MPN, first Most Probable Number (MPN), please, then just use MPN. Which class is the 260 MPN?
L. 67: Why oysters can not be both “reef-builders “and” as filter-feeders”? After all, they are filter-feeders.
MM
L. 83-85: This is relevant information that should be placed in the introduction section.
L. 87-88: How long it took from sampling site to the laboratory?
L. 88: What “clean seawater” means? Artificial seawater or sterile seawater?
L. 90-95: Overall, this part needs a lot of clarifications. The edible content comprises of the three portions or parts “flesh, hemolymph and intra-valvular liquid)”. So, the analyses were performed for the pooled samples or each part was analyzed separately, please clarify? Wait a minute, hemolymph of 10 individuals were used for metagenomic analysis, correct? However, the other five pools were also subjected to metagenomic analysis and contained hemolymph as well, is it correct? Please change “metagenomic” to “metabarcoding”. Additionally, what is/are the difference(s) between microbiological analysis, and HAV, HEV and norovirus that are also microorganisms and why the latter three were only investigated in the digestive gland? The same is true for the edible portions, why Salmonella and Listeria were not part of the microbiological analysis as well as OsHV-1? This section also misses the analysis performed with farming waters. Figure 1 needs better resolution for publication.
L. 105: “The microbiological analysis of flesh and intra-valvular liquid samples” it is confusing especially considering figure 1. How was it done, flesh of the edible portions was analyzed alone or together with intra valvular liquid and hemolymph? Please, clarify. Also, farming waters were also subjected to microbiological analysis, but they were not present in this section.
L. 108-115: Those are the analyses required by the European Union microbiological criteria, correct? For Salmonella and Listeria only the detection was required? How it was done, with specific media? Please, provide more details so any of your readers can repeat the analyses.
L. 117-123: I am sure that for the majority of your readers all those codes do no mean anything. Please, provide more details. It can be in supplementary material. Remember that anybody that want to perform the same analyses would have all the information needed in a single manuscript.
L. 126-227: “incorporated” or “plated”. Incorporated sounds like the 1 ml were mixed with the medium prior plating.
L. 127-128: For Pseudomonas, 100 uL of each sample was used, without serial dilution? For enumeration of Pseudomonas, it would be more reliable if a serial dilution was performed.
L. 141-144: Please cite figure 1, because it is the first time that any analysis is performed with the farming waters that I am assuming they are the same as filtering waters. In figure 1 the microbiological analysis performed with farming waters were actually the antimicrobial screening? I guess not, because it is mentioned that E. coli and Enterococcus sp. were isolated, please clarify it. Also, as far as I can tell Kirby-Bauer method and disk diffusion method are exactly the same technique, right? Whether so, please provide just one name, otherwise it can be quite confusing.
L. 145-156: I believe all the information regarding the antimicrobials would be better in a table, where the repetition can be avoided and it would be a lot easier to understand.
L. 161-165: Please, cite figure 1. Could the authors briefly describe how the analysis was done. Why it was only performed in the farming waters and digestive gland? Overall, a better explanation is needed on why a certain analysis was performed in one or more type of samples and not on other parts. Also, all the analyses performed so far were done for all for seasons or not?
L. 167-175: Again, why metabarcoding was only performed in flesh samples? Are the authors sure about the primer pair used? I have never seen this fragment size being sequenced with NGS. Chimeras was not the only parameters that should be checked in the dataset? Besides, with the information provided nobody can repeat this analysis. Please, provide more information, which pipeline and paraments were used? It can be provided in a supplementary material. Every analysis performed with NGS is done by computer analysis, so it does not mean anything! How about statistics analyses, were they performed? how?
Results
L. 181-182: According to the graph, the lowest weight was obtained in spring, please verify.
L. 205: Please use one single term to refer to the waters, farming, filtering or growing, otherwise, it is quite hard do follow.
L. 205-208: It is discussion. In this section, please provide only the results obtained.
L. 210-211: “clearly below the legal limit for E. coli contamination in oysters reared 210
in a class B area (> 230 - ≤ 4600 NMP E. coli/100 g).” it is discussion. Here just provide the count number, please. “(> 230 - ≤ 4600 NMP E. coli/100 g).” it is relevant information to include in the introduction as well. Please, change “NMP” to “MPN”.
L. 212-214: Please refer to table 1, where the results are shown. Salmonella was not detected in autumn and spring.
L. 216: Please change “microorganisms” to “bacterial” or “virus”, because only two types of microorganisms were investigated.
L. 218-220: Please revise those results. For instance, in summer the values were similar, but with two logarithmic.
L. 225-226: Not for winter and spring, please revise.
L. 239-245: It means that counting below 100 cells were not considered? Whether so, E. coli in 75% of the samples is incorrect, because it is only present in 5 of the 12 counting’s. Please, explain it and fix if needed.
L. 295-296: And winter as well for water and edible portion, please verify.
L. 317-318: “NoV was detected in the digestive gland in spring and summer samples, as well as in the farming water in spring.” according to table 5, NoV was present in the digestive gland in autumn, spring and summer and not detected in any other sample. Please verify.
L. 330-331: How hemolymph contributes to the abundance of Vibrio??? Besides, it is discussion.
Legend of Figure 6 and note Table 6, please provide the meaning of the acronyms in the same way they appear in the figure and table, please. Still in figure 6, please remove the “z” from “z_OTHER”, please.
Discussion
L. 345-348: Some of this description could be used in the introduction section, where the authors explain what they did in the present study and it would be nice to have a characterization of the sampling place.
L. 351-353: How rain can be responsible for the detection of pathogenic bacterial species during winter?
L. 358-359: I do not agree with Pseudomonas, in every season the detection was < 100. Please have a look into it.
L. 367-378: Remember the Salmonella was present in the farming water in summer and winter. Even though with low counting, enterococcus was also found in all seasons.
L. 388-390: How those bacteria could affect the human body after consumption, as all have pathogenic species.
L. 390-391: How species specific are the FISH probes to assure that these species were the only ones that appear?
L. 405-407: I did not understand the relationship here.
L. 412: Afterwards, the sampling occurred at Lagoa de Mira of Canal de Mira?
L. 412-414: According to table 5, only NoV occurred.
L. 429-431: Not only E. coli, it also holds true for Enterococcus.
Author Response
Reviewer 5
Comments and Suggestions for Authors
Abstract
- 28-29: what does Class B production area mean?
According to European Union legislation (Regulation (EU) 2019/627), the requirements for class B areas are: i) live bivalve molluscs may be collected and placed on the market for human consumption only after treatment in a purification centre or after relaying; ii) >230 - ≤ 4600 MNP E. coli per 100 g of flesh and intravalvular liquid. This information has now been included in the manuscript as follows: “The level of E. coli contamination was clearly below the legal limit for E. coli contamination in oysters reared in a class B area (> 230 - ≤ 4600 NMP E. coli/100 g).”
Introduction
- 58-66: How the monitoring is done? How frequently the monitoring occurs? How many oysters are usually used for it? For the broad audience this classification has to be better explained. Is it based on NMP of E. coli? What are the limits of NMP for Class A, B and C? Additionally, change the order of MPN, first Most Probable Number (MPN), please, then just use MPN. Which class is the 260 MPN?
The competent authorities shall classify production and relaying areas from which they authorise the harvesting of live bivalve molluscs as Class A, Class B and Class C areas according to the level of faecal contamination of flesh and intravalvular liquid. To classify production and relaying areas the competent authorities shall fix a review period for sampling data from each site. The classification and its requirements have already been described the in introduction (“European Hygiene Regulations [14–16] state that shellfish business operators are responsible for ensuring that bivalve molluscs meet strict hygiene and health standards. Risk assessment and management currently rely on the classification of shellfish harvesting areas based on the results of monitoring E. coli in shellfish [17] as an indicator of faecal contamination. Depending on shellfish production area classification (A, B or C), oysters with less than 230 MPN (Most Probable Number) of E. coli per 100 g of flesh and intra-valvular liquid may go to market for direct human consumption. Nevertheless, those harvested from Class B (> 230 - ≤ 4600 MPN E. coli/100 g) may be collected and placed on the market for human consumption only after treatment in a purification centre or after relaying and oysters harvested from class C areas (less than 46,000 MPN of E. coli/100g) must be submitted for relaying over a long period or undergo heat treatment to eliminate pathogenic microorganisms before being sold to consumers [16].”).
- 67: Why oysters can not be both “reef-builders “and” as filter-feeders”? After all, they are filter-feeders.
Indeed, oysters are both reef-builders and filter-feeders. It was corrected as it follows “oysters are keystone species in estuarine environments either as reef-builders and as filter-feeders”.
MM
- 83-85: This is relevant information that should be placed in the introduction section.
We corrected this by adding it to the introduction (study objective).
- 87-88: How long it took from sampling site to the laboratory?
The time interval between sampling and processing in the laboratory did not exceed 3 hours, and the samples were transported in temperature-controlled food boxes. This information was added to the text: “Samples were transported within 3 hours in temperature-controlled food boxes”.
- 88: What “clean seawater” means? Artificial seawater or sterile seawater?
We thank the reviewer for the question. Indeed, clean seawater corresponds to seawater which is free from microbiological and chemical contamination. This was corrected in the text.
- 90-95: Overall, this part needs a lot of clarifications. The edible content comprises of the three portions or parts “flesh, hemolymph and intra-valvular liquid)”. So, the analyses were performed for the pooled samples or each part was analyzed separately, please clarify? Wait a minute, hemolymph of 10 individuals were used for metagenomic analysis, correct? However, the other five pools were also subjected to metagenomic analysis and contained hemolymph as well, is it correct? Please change “metagenomic” to “metabarcoding”. Additionally, what is/are the difference(s) between microbiological analysis, and HAV, HEV and norovirus that are also microorganisms and why the latter three were only investigated in the digestive gland? The same is true for the edible portions, why Salmonella and Listeria were not part of the microbiological analysis as well as OsHV-1? This section also misses the analysis performed with farming waters. Figure 1 needs better resolution for publication.
We agree that the resolution of Figure 1 should be improved and have reloaded all the figures with higher quality. Figure 1 (now Figure 2) was reformulated to better illustrate the study design, the spectrum of samples studied, and the methods used. Additionally, more detailed information was added to the materials and methods: “For bacterial enumeration, a pooled sample comprising 25 oysters was set. With regard to the detection of Salmonella spp. and L. monocytogenes, five pools comprising five oyster each were prepared. Total aerobic microorganisms at 30 °C, marine heterotrophic bacteria at 21 °C, E. coli and Enterococcus spp. were also assessed on the superficial biofilm, intra-valvular liquid and hemolymph samples (pool of 10 oysters.”
We accept the reviewer’s suggestion to change the word metagenomic to metabarcoding.
HAV and norovirus were investigated in the digestive gland following the guidelines of ISO/TS 15216-1:2013 ‘Microbiology of food and animal feed — Horizontal method for determination of hepatitis A virus and norovirus in food using real-time RT-PCR — Part 1: Method for quantification’. HEV, also a (+)ssRNA virus, was included in this analysis since there is no specific guideline for HEV quantification in this type of food. Moreover, viral particles bind specifically to digestive ducts making the digestive gland the primary target organ to bioaccumulate these viruses (Le Guyader et al, 2006).
- 105: “The microbiological analysis of flesh and intra-valvular liquid samples” it is confusing especially considering figure 1. How was it done, flesh of the edible portions was analyzed alone or together with intra valvular liquid and hemolymph? Please, clarify. Also, farming waters were also subjected to microbiological analysis, but they were not present in this section.
According to the legislation (Commission Regulation 2073/2005 on microbiological criteria for foodstuffs) the edible portion of oysters and other live bivalve molluscs encompasses flesh and intra-valvular liquid (“The microbiological analysis of flesh and intra-valvular liquid samples was performed in compliance with the European Union microbiological criteria for live bivalve molluscs…”-2.2 section), while hemolymph corresponds to the circulatory fluid of bivalves. This vascular fluid, which is able to trigger a wide range of defense mechanisms, transports nutrients, respiratory gases, enzymes, metabolic wastes, and toxicants throughout the body of oysters. This information was added in the introduction section.
The microbiological analysis of farming waters samples is mentioned in the final of the first paragraph of section 2.2 “Finally, the total counts of aerobic microorganisms at 22 °C and 37 °C, marine hetero-trophic bacteria at 21 °C, E. coli, Enterococcus spp., and Salmonella spp. were also evaluated in the farming water samples”.
- 108-115: Those are the analyses required by the European Union microbiological criteria, correct? For Salmonella and Listeria only the detection was required? How it was done, with specific media? Please, provide more details so any of your readers can repeat the analyses.
The European Commission only requires the enumeration of E. coli and the detection of Salmonella for molluscan shellfish. The other microorganisms have been included for their importance to food safety and following other guidelines.
Regarding the detection of Salmonella spp. and Listeria monocytogenes, it was performed according to ISO 6579 and ISO 11290-1, respectively, which includes specific medium, incubation temperatures and revitalisation medium. Since the methods are extensively described in each ISO, the authors believe it will overload the Materials and Methods section.
More details about the methodology used are provided in Table S1 (which we have added now).
- 117-123: I am sure that for the majority of your readers all those codes do no mean anything. Please, provide more details. It can be in supplementary material. Remember that anybody that want to perform the same analyses would have all the information needed in a single manuscript.
More details, such as the selective media and temperature of incubation used, are provided as supplementary information in a summary table (now Table S1).
- 126-227: “incorporated” or “plated”. Incorporated sounds like the 1 ml were mixed with the medium prior plating.
We accept the reviewer’s suggestion.
- 127-128: For Pseudomonas, 100 µL of each sample was used, without serial dilution? For enumeration of Pseudomonas, it would be more reliable if a serial dilution was performed.
Serial dilutions were performed so that the results would be more reliable. It was corrected in the text: “The detection of Pseudomonas spp. was performed using serial dilutions and spreading 100 µl of each sample on Cephaloridin Fucidin Cetrimide (CFC) agar…”.
- 141-144: Please cite figure 1, because it is the first time that any analysis is performed with the farming waters that I am assuming they are the same as filtering waters. In figure 1 the microbiological analysis performed with farming waters were actually the antimicrobial screening? I guess not, because it is mentioned that E. coli and Enterococcus sp. were isolated, please clarify it. Also, as far as I can tell Kirby-Bauer method and disk diffusion method are exactly the same technique, right? Whether so, please provide just one name, otherwise it can be quite confusing.
The farming waters (filtering water or water column) are mentioned in section 2.2 Bacterial analysis, where the microbiological analysis and the microorganisms studied are described: “the total counts of aerobic microorganisms at 22 °C and 37 °C, marine heterotrophic bacteria at 21 °C, E. coli, Enterococcus spp., and Salmonella spp. were also evaluated in the farming water samples.”
Antimicrobial susceptibility testing was performed only for Enterococcus spp. and E. coli isolated from all samples (farming water, edible portion, intra-valvular liquid, superficial biofilm and hemolymph) by the Kirby-Bauer method. The Kirby-Bauer method is the same as the diffusion method. We have removed the second term in order not to confuse the readers.
- 145-156: I believe all the information regarding the antimicrobials would be better in a table, where the repetition can be avoided and it would be a lot easier to understand.
The authors agree with the suggestion and a table was created (now Table 1).
- 161-165: Please, cite figure 1. Could the authors briefly describe how the analysis was done. Why it was only performed in the farming waters and digestive gland? Overall, a better explanation is needed on why a certain analysis was performed in one or more type of samples and not on other parts. Also, all the analyses performed so far were done for all for seasons or not?
HAV and norovirus were investigated in the digestive gland and farming water following the guidelines of ISO/TS 15216-1:2013 ‘Microbiology of food and animal feed — Horizontal method for determination of hepatitis A virus and norovirus in food using real-time RT-PCR — Part 1: Method for quantification’. The digestive gland was the chosen oyster compartment to search for these viruses as recommended by the guideline and widely used by other research groups. On top of that, viral particles bind specifically to digestive ducts making the digestive gland the primary target organ to bioaccumulate these viruses (Le Guyader et al, 2006). Details on the virus analysis were added to the manuscript.
NoV, HAV and HEV detection and quantification were performed in farming water and digestive gland samples from all seasons. OsHV-1 detection and quantification was performed in oyster edible portion samples also from all seasons.
- 167-175: Again, why metabarcoding was only performed in flesh samples? Are the authors sure about the primer pair used? I have never seen this fragment size being sequenced with NGS. Chimeras was not the only parameters that should be checked in the dataset? Besides, with the information provided nobody can repeat this analysis. Please, provide more information, which pipeline and paraments were used? It can be provided in a supplementary material. Every analysis performed with NGS is done by computer analysis, so it does not mean anything! How about statistics analyses, were they performed? how?
Metabarcoding of flesh samples and individual compartments (intra-valvular liquid, superficial biofilm and hemolymph) of the oyster was planned. Unfortunately, only the flesh samples (from all seasons) and two samples of hemolymph had enough DNA to go be subjected to the analysis.
All the pipelines and analyses were designed and developed by GATC Biotech from Eurofins Genomics. The data analysis was also performed by the company.
A detailed explanation of the metagenomic data analysis and the softwares used was added to the 2.5 section.
Results
- 181-182: According to the graph, the lowest weight was obtained in spring, please verify.
We apologise for this misunderstanding. It was amended as follows “Total weight varied between 5670.97 g ± 57.09 (spring; mean body weight ± S.D.)”.
- 205: Please use one single term to refer to the waters, farming, filtering or growing, otherwise, it is quite hard do follow.
We agree that using more than one term to define the same sample can be confusing. Therefore, we have substituted all the other terms used before for farming waters throughout the text.
- 205-208: It is discussion. In this section, please provide only the results obtained.
That information was removed from the results and moved to the Discussion as follows “Regardless of the sampling period, the edible portion of oysters showed compliance with the microbiological safety criteria set out in [16,17,28–31]. The level of E. coli contamination was clearly below the legal limit for E. coli contamination in oysters reared in a class B area (> 230 - ≤ 4600 NMP E. coli/100 g)”.
- 210-211: “clearly below the legal limit for E. coli contamination in oysters reared in a class B area (> 230 - ≤ 4600 NMP E. coli/100 g).” it is discussion. Here just provide the count number, please. “(> 230 - ≤ 4600 NMP E. coli/100 g).” it is relevant information to include in the introduction as well. Please, change “NMP” to “MPN”.
It was rephrased as “the level of E. coli contamination was found between 20 (summer and spring) and 92 (winter) MPN E. coli/100 g in the edible portion” and the interval of the legal limit of E. coli was replaced in both the introduction and the discussion.
- 212-214: Please refer to table 1, where the results are shown. Salmonella was not detected in autumn and spring.
The reference to the Table 1 (now Table 2) was in the first paragraph of section 3.2 (“In the present study, the microbiological quality of oysters and their growing farming waters was examined in four seasonal sampling surveys (Table 2)”. We agree that the result of Salmonella spp. in farming water during winter and summer should be added and we added more information (“and Salmonella spp. was only detected in summer and winter in farming water samples.”).
- 216: Please change “microorganisms” to “bacterial” or “virus”, because only two types of microorganisms were investigated.
“The number of total microorganisms” refers to the total number of aerobic microorganisms, which include cultivable bacteria, molds and yeasts.
- 218-220: Please revise those results. For instance, in summer the values were similar, but with two logarithmic.
We apologise for this misunderstanding. It was corrected as follows “…while similar values were found in autumn, in summer and in spring the difference exceeded two logarithms”
- 225-226: Not for winter and spring, please revise.
We apologise for this misunderstanding. It was corrected as follows “…A higher concentration of microorganisms in the intra-valvular liquid was found compared to the farming water column in summer and autumn seasons.”
- 239-245: It means that counting below 100 cells were not considered? Whether so, E. coli in 75% of the samples is incorrect, because it is only present in 5 of the 12 counting’s. Please, explain it and fix if needed.
The detection limit of the FISH technique is 100 cells/ml. Therefore, if we do not detect any cells by this method, we can only infer that the quantity of that bacteria in the sample is below the detection limit of the technique (100 cells/ml).
What the authors intend to convey with the phrase: “The FISH method allowed the detection of E. coli cells in 75% of the superficial biofilm samples (summer, winter and spring)” is that E. coli was recovered from superficial biofilm samples (n=4) in summer, winter and spring samples (n=3; 75%), but not from the autumn sample.
- 295-296: And winter as well for water and edible portion, please verify.
We apologise for this misunderstanding. It was amended as follows: “…Regarding seasonality, MDR E. coli were found in farming water in summer, autumn and winter samples and the edible content in autumn and winter samples.”
- 317-318: “NoV was detected in the digestive gland in spring and summer samples, as well as in the farming water in spring.” according to table 5, NoV was present in the digestive gland in autumn, spring and summer and not detected in any other sample. Please verify.
We apologise for this misunderstanding. The description of the results in the text is correct, so we have corrected the information in the Table.
- 330-331: How hemolymph contributes to the abundance of Vibrio??? Besides, it is discussion.
This information and a possible explanation are presented in the discussion section: “Moreover, this study showed that hemolymph contained more Vibrio spp. compared to the edible content, which could be explained by the immunological function of hemolymph. Indeed, the overall microbiome of oysters displays a seasonal influence and similar conclusions were reached by Scannes et al. (2021) [54].”
Therefore, the sentence “In addition, the hemolymph seemed to have a larger contribution to the amount of Vibrio spp. compared to the edible content.” was removed from the results section.
Legend of Figure 6 and note Table 6, please provide the meaning of the acronyms in the same way they appear in the figure and table, please. Still in figure 6, please remove the “z” from “z_OTHER”, please.
The notes on Table 6 (now Table 7) were amended and the “z” was removed from Figure 6 (now Figure 7).
Discussion
- 345-348: Some of this description could be used in the introduction section, where the authors explain what they did in the present study and it would be nice to have a characterization of the sampling place.
The authors agree that some information about the sampling location is missing. Therefore, the information has been added in the final paragraph of the introduction: “Commercial oysters included in this study were farmed in Canal de Mira, one of the leading producers in Portuguese western coast, that receives a continuous seawater and freshwater supply, but also inland drainage and treated and untreated urban wastewater.”
- 351-353: How rain can be responsible for the detection of pathogenic bacterial species during winter?
Following this sentence, we have cited Liu et al, 2018, that using evidence from their research on the presence and persistence of Salmonella spp. in water, stated that “Salmonella can be carried to surface waters through rainfall and surface runoffs, survive many challenges such as ultraviolet (UV) radiation from sunlight, poor nutrients, the changes in pH, and temperature”.
- 358-359: I do not agree with Pseudomonas, in every season the detection was < 100. Please have a look into it.
Indeed, Pseudomonas spp. was <100 in all seasons, as we have amended as “samples collected in the rainy seasons of autumn and winter showed the highest total microorganisms and C. perfringens contamination.”
- 367-378: Remember the Salmonella was present in the farming water in summer and winter. Even though with low counting, enterococcus was also found in all seasons.
Despite the presence of Salmonella spp. and Enterococcus spp. in the farming water, the legal bacteriological requirements of the edible portion were accomplished. Therefore, these oysters were satisfactory for human consumption after placement in depuration or relaying areas (production in Class B area). Moreover, Salmonella spp. was not detected in the edible portion.
- 388-390: How those bacteria could affect the human body after consumption, as all have pathogenic species.
All animals may have potentially pathogenic bacterial species. These bacteria only cause disease if (i) they are in the minimum number necessary to cause disease under normal conditions (infecting dose) or (ii) if the host (the person who consumes the contaminated animal) has depressed immune defences (immunosuppressed).
- 390-391: How species specific are the FISH probes to assure that these species were the only ones that appear?
16S rRNA oligonucleotide probes were described in material and methods section. However, the in silico specificity of the probes was not confirmed by BLASTn or ProbeCheck searches because they were previously used by us and others. In each assay, the universal Bacteria probe, Eub338 (5´-GCTGCCTCCCGTAGGAGT-3´), and a nonsense probe, Non338, were also used.
- 405-407: I did not understand the relationship here.
We understand that this may not be clear. The presence of Vibrio spp. has a dual importance, as this bacterium may be either an oyster pathogen (Vibrio anguillarum) or a food born pathogen (V. parahaemolyticus and V. vulnificus). So, we have rephrased the sentence as follows: “Vibrio spp. plays an important role in oyster welfare, but also in public health, since it could be either an oyster pathogen, associated with the mass mortality of Crassostrea gigas, or a zoonotic pathogen, including V. parahaemolyticus (the principal causes of seafood borne disease linked to the consumption of shellfish) and V. vulnificus, that may cause fulminant wound infections.”
- 412: Afterwards, the sampling occurred at Lagoa de Mira of Canal de Mira?
We thank the reviewer for the comment. Indeed, “Canal de Mira” and “Lagoa de Mira” name the same body of water, which is quite long and shallow and communicates with the Ria de Aveiro through a narrow isthmus (which is only kept open to hydrodynamic circulation through dredging). As such, some consider it a channel “Canal” while others a lagoon “Lagoa”. We have replaced the term “Lagoa de Mira” with “Canal de Mira” to be less confusing.
- 412-414: According to table 5, only NoV occurred.
NoV was detected in the digestive gland in spring and summer samples, as well as in the farming water in spring. HEV was detected in the column farming water in spring. Table 7 was corrected accordingly.
- 429-431: Not only E. coli, it also holds true for Enterococcus.
We agree with the Referee and have rephrased it as follows “…as evidenced by the isolation of both E. coli and Enterococcus spp. multidrug-resistant strains and the high frequency of resistance towards important classes of antimicrobial drugs”.
Round 2
Reviewer 2 Report
The manuscript had a great improvement after the present revision. It´s almost a new article. Congratulations for that.
Just one minor suggestion at the introduction. In lines 54-56, the authors must change the phrase, because the human illness cases are not only associated to exposure to contaminated sellfish. Try to use “… can be associated to exposure to contaminated shellfish.”
Author Response
The manuscript had a great improvement after the present revision. It´s almost a new article. Congratulations for that.
Just one minor suggestion at the introduction. In lines 54-56, the authors must change the phrase, because the human illness cases are not only associated to exposure to contaminated sellfish. Try to use “… can be associated to exposure to contaminated shellfish.”
The manuscript had a great improvement after the present revision. It´s almost a new article. Congratulations for that.
Just one minor suggestion at the introduction. In lines 54-56, the authors must change the phrase, because the human illness cases are not only associated to exposure to contaminated sellfish. Try to use “… can be associated to exposure to contaminated shellfish.”
We would like to thank Referee 1 for the positive comment on the revised version of the manuscript.
We thank the Referee suggestion. We have now rephrased the paragraph as it as follows: “Other important zoonotic agents, such as hepatitis A virus (HAV), hepatitis E virus (HEV), Escherichia coli (E. coli), and Clostridium spp. can be associated to exposure to contaminated shellfish [10–12].”
Reviewer 4 Report
I have no further comments.
Author Response
I have no further comments.
We would like to thank Referee 4 for the positive comment on the revised version of the manuscript.